# Compact 5G Nonuniform Transmission Line Interdigital Bandpass Filter for 5G/UWB Reconfigurable Antenna

**DOI:** 10.3390/mi13112013

**Published:** 2022-11-18

**Authors:** Sahar Saleh, Mohd Haizal Jamaluddin, Faroq Razzaz, Saud M. Saeed

**Affiliations:** 1Wireless Communication Centre, Universiti Teknologi Malaysia (UTM), Johor Bahru 81310, Malaysia; 2Department of Electronics and Communications Engineering, Faculty of Engineering, Aden University, Aden 5243, Yemen; 3Electrical Engineering Department, College of Engineering, Prince Sattam Bin Abdulaziz University, Al-Kharj 16278, Saudi Arabia; 4Faculty of Engineering and Information Technology, Taiz University, Taiz 6803, Yemen

**Keywords:** 5G, interdigital bandpass filter (IBPF), nonuniform transmission lines (NTLs) theory, uniform transmission line (UTL), HFSS

## Abstract

In this study, at two different fifth generation (5G) low-frequency bands (3.7–4.2 GHz and 5.975–7.125 GHz) and based on nonuniform transmission lines (NTLs) theory, a compact three-quarter-wave resonators interdigital bandpass filter (IBPF) is analyzed, designed, and fabricated. The compact proposed filter is considered as a good candidate for reconfigurable 5G low-frequency bands and ultrawide band (UWB) antenna, which will reduce the size of the final RF communication system. Firstly, a uniform transmission line (UTL) IBPF at these two bands is designed and tested; then the NTL concept is applied for compactness. For both UTL and NTL IBPFs, different parametric studies are performed for optimization. At the first frequency band, size reductions of 16.88% and 16.83% are achieved in the first (symmetrical to the third resonator) and second λ/4 resonator of UTL IBPF, respectively, with up to 36.6% reduction in the total area. However, 16.46% and 16.33% size reductions are obtained in the first (symmetrical to the third resonator) and second λ/4 resonator, respectively, at the second frequency band with a 40.53% reduction in the whole circuit area. The performance of the proposed NTL IBPF is compared with the UTL IBPF. The measured reflection coefficient of the proposed NTL IBPF, S_11_, appears to be less than −10.53 dB and −11.27 dB through 3.7–4.25 GHz and 5.94–7.67 GHz, respectively. However, the transmission coefficient, S_12_ is around −0.86 dB and–1.7 dB at the center frequencies, *f_c_* = 3.98 GHz and 6.81 GHz, respectively. In this study, simulations are carried out using high-frequency structure simulator (HFSS) software based on the finite element method (FEM). The validity of the proposed theoretical schematic of this filter is proved by design simulations and measured results of its prototype.

## 1. Introduction

Bandpass filters (BPFs) are mainly used in many RF/microwave applications to control the required frequency band [1]. The most popular compact structure BPFs are the interdigital and hairpin BPFs (HPBF and IBPF) [2,3]. Many techniques are applied to HPBF for compactness and performance enhancement such as using defected ground structures (DGS) [4], defected microstrip structures (DMS) [5], artificial left-handed and right-handed transmission lines (LHRHTLs) [6], multilayers techniques [7], high dielectric substrates [8], and nonuniform transmission lines (NTLs) [9,10]. However, HPBF suffers from high-order harmonics, which is not suitable for reconfigurable 5G narrowband and UWB antenna applications such as cognitive radio networks (CRNs). IBPFs are preferred over HPBFs in modern wireless communication systems due to their excellent performance, wide passband, spurious second harmonic suppressions, and compact size of λ/4 resonators. The problem of second harmonics in HPBFs is resolved in this work using IBPF. IBPF is designed and used in many applications at different frequency bands [8,11,12,13,14,15,16]. Further harmonics suppression in the IBPF response is achieved using different techniques such as etching DGS to the ground [17], etching spurlines on the resonators [18], and the use of under-coupled quarter-wavelength resonators [19]. Several size-reduction technologies are used to reduce the circuit area of the IBPF. Low-temperature co-fired ceramic (LTCC) technology is employed in [20] to design a compact ISM band IBPF. The authors in [21] designed a compact 2.8 GHz IBPF based on a composite right–left-handed (CRLH) cell resonator using an improved interdigital capacitor combined with a meander electromagnetic band gap (EBG). Step impedance resonators (SIRs) were used to reduce the size of Ku- band IBPF, which was fabricated using a silicon-based substrate on micro-electro-mechanical systems (MEMS) in [22]. Compact UWB IBPF was designed in [23] using multilayer liquid crystal polymer technology (LCPT). Moreover, in [24,25], compact inject-printed IBPFs were designed using LCPT. As compared to the conventional end-coupled BPF, a 60% size reduction was obtained in [26] using a modified IBPF with alternate λ/4 and λ/2 resonators to obtain a narrow X-band (9.85–10.3 GHz). In [27], a compact X-band IBPF was designed using an ultra-thin LCP substrate. Choosing a high dielectric substrate helps in size reduction as in [8,11,17,18]. A compact K-band 20.4 GHz IBF was designed using two layers of silicon substrate based on MEMS technology in [16]. The authors in [28] designed a compact 0.8 GHz IBPF using a spiral and folded SIR. However, in [29], a high selectivity IBPF with 10 controllable transmission zeros (TZs) was proposed using square complementary split-ring resonators with open-circuited stubs. Recently, millimeter-wave (mmW) IBPFs based on integrated passive devices (IPDs) were designed in [30,31]. In [30], through-quartz vias (TQVs) were added to the proposed IBPF in [31] for coupling adjustment and size reduction. However, for IBPF’s compactness, wideband, and better harmonics suppressions, through-silicon via (TSV)-based 3-D integrated circuit (3-D IC) technology was used in [30].

To avoid the fabrication difficulties and high cost in designing the required compact IBPF, NTLs theory [9,10,32,33,34] is applied for the first time in this paper to reduce its size at two 5G low-frequency bands proposed by the federal communication commission (FCC): licensed (C-Band: 3.7–4.2 GHz) and unlicensed (5.975–7.125 GHz) [35]. Although the NTLs concept was previously applied to HPBF in [9,10], resulting in a 17.79% size reduction in its λ/2 resonators, the reduction in the total circuit area is not significant as compared to the results reported in this work. Due to the importance of UWB technology [36,37,38,39,40,41,42,43] and 5G low-frequency bands [10,44,45,46,47] in the modern wireless communication system in terms of high data rate, compatibility with consumer demand, and low latency, switching between these bands is more practical in terms of size and time. For this purpose, as shown in Figure 1, the designed filter will be utilized as a part of our future project that aims to design a compact reconfigurable UWB/5G low-frequency band antenna suitable for cognitive radio network (CRN) applications. Some recent works using this concept can be found in [48,49,50,51,52]. The chosen substrate material is Rogers RO4003C (Ɛ𝑟 = 3.55, tanδ = 0.0027, and h = 0.813 mm) and the copper thickness is 0.035 mm. The remainder of the paper is organized as follows: Section 2 explains the NTLs theory and how it is applied to the filter’s resonators. Section 3 explains the methodology of the work, while Section 4 demonstrates the results and discussions.

## 2. Nonuniform Transmission Lines (NTL) Theory

Compact-size passive microwave components that are more effective and less expensive are crucial for the compatibility with current industrial needs in the modern wireless communication system. Recently, NTLs theory [9,10,32,33,34] has been widely used in many microwave components to reduce their size by reducing the length of their conventional uniform transmission lines (UTLs) at different percentage levels. Using this theory, compactness is achieved when UTL with length *d_0_*, constant characteristics impedance *Z_0_*, and propagation constant *β* is replaced by the equivalent NTL with smaller length *d* (*d* < *d_0_*), varying characteristics impedance *Z*(*z*), and propagation constant *β*(*z*). The performance of these lines will be equivalent if their *ABCD* parameters are equal. Both UTL and NTL are demonstrated in Figure 2.

The ABCD parameters of UTL are
(1)$$\left[\begin{array}{cc} A_{0} & B_{0}\\ C_{0} & D_{0} \end{array}\right]\;=\;\left[\begin{array}{cc} \cos\left(\theta_{0}\right) & jZ_{0}\sin\left(\theta_{0}\right)\\ \frac{j}{Z_{0}}\sin\left(\theta_{0}\right) & \cos\left(\theta_{0}\right) \end{array}\right]$$
where $\theta_{0}$ is the electrical length ($\beta_{0\;}d_{0}$) of the UTL.

To obtain the *ABCD* parameters of NTL, it is subdivided into *K* UTLs as illustrated in Figure 2b and then their ABCD parameters multiplied to obtain the total ABCD matrix as follows
(2)$$\left[\begin{array}{cc} A & B\\ C & D \end{array}\right]=\prod_{i=1}^{K}\left[\begin{array}{cc} A_{K} & B_{K}\\ C_{K} & D_{K} \end{array}\right]$$
where *Ai* = *Di* = cos (Δ*θ*), Bi = j Z(z) ((*i* − 0.5)Δ*z*$)\;\sin\left(\Delta \theta \right)$, Ci = $\frac{\;\mathrm{jsin}\;\left(\Delta \theta \right)}{Z\;\left(\left(i-0.5\right)\Delta z\right)\;}$, *i* = 1, 2… K, Δ*z* = d/K, and Δθ = $\frac{2\pi }{\lambda }\Delta z=\frac{2\pi f}{c}\sqrt{\varepsilon_{eff}}\;\Delta z$. Here, c is the speed of light and $\varepsilon_{eff}$ is the effective dielectric constant.

The NTL’s *Z*(*z*) can be expanded in a truncated Fourier series as follows
(3)$$\;\ln(Z\left(z\right)/Z_{0})\;=\;\sum_{n=0\;}^{N}C_{n}\;\cos(\frac{2\pi nz\;}{d})$$

The number of coefficients N is chosen to be 10 for more convergence [53]. The Fourier series coefficients, *C_n_*s, are optimized using a built-in MATLAB function known as “fmincon” to minimize the error function given by (4) through the frequency bands (3.7–4.2 GHz) and (5.975–7.125 GHz) to have the performance equivalency between UTL and NTL
(4)$$\mathrm{Error}=\sqrt{\frac{1}{M}\sum_{m=1}^{M}\frac{1}{4}\left(\left|A-A_{0}{|}^{2}+Z_{0}^{-2}\right|B-B_{0}{|}^{2}+Z_{0}^{2}\left|C-C_{0}{|}^{2}+\right|D-D_{0}{|}^{2}\right)}$$
where *M* is the number of the frequencies *f_m_* (*m* = 1, 2,…*M*) within the desired band with frequency increment Δ*f* and *A*_0_, *B*_0_, *C*_0_, and *D_0_* are the *ABCD* matrix parameters of UTL defined in (1), respectively. Two constraints should be considered to restrict the error function in (4) such as physical matching: $\bar{Z}$(0) = $\bar{Z}$(d) = 1 (NTL and UTL should have the same widths at the two ends) and easy fabrication with allowable minimum and maximum widths: $\bar{Z}$_min_$\;\leq \bar{Z}\left(z\right)\leq \;\bar{Z}$_max_, where $\bar{Z}$_min_ and $\bar{Z}$_max_ are the minimum and the maximum normalized characteristic impedances of the UTL, respectively. According to the study carried out in [9,10] on designing the NTL HPBF, the width of the NTL resonator should be between the UTL filter’s resonator width (*W_res_*) and the lowest allowed width for fabrication, *W_min_*= 0.3 mm. In this work, using NTLs theory and based on these constraints, each UTL resonator of IBPF was replaced with its equivalent NTL resonator.

## 3. Design of The Proposed Compact NTL 3.95 and 6.55 GHz IBPFs

To compare the compactness achieved in the NTL IBPF at both bands, a UTL IBPF was firstly designed, then NTLs theory was applied for compactness. UTL IBPF consists of an array of *n* open-circuited λ/4 resonators, which are also short-circuited at the other end in alternative orientation as shown in Figure 3a, where *l_i_*’s, *W_i_* ‘s, *θ_t_*, *Y*_1_ =*Y_n_*, *S*_*i*,*i*+1_ (*i* = 1 to *n*) are the resonators’ lengths, resonators’ widths, electrical length for tapping position, input (=output) characteristics admittances and the space between adjacent resonators, respectively. The equivalent circuit of IBPF is shown in Figure 3b, where *L_i_*s, *C_i_*s, *C*_*i*,*i*+2,_ and *C_t_* are the resonator’s inductance, the resonator’s capacitance, the coupling capacitance between resonators, and the compensate capacitance for tapping, respectively. According to [1], the design equations for symmetrical coupled lines IBPF are
(5)$$\theta =\frac{\pi }{2\;}\;\left(1-\frac{FBW}{2}\;\right),Y=\frac{Y_{1}}{\tan\left(\theta \right)},\;J_{i,i+1}=\frac{Y}{\sqrt{g_{i}g_{i+1}}}\;,\;\mathrm{for}\;i=1\;\mathrm{to}\;n-1,\;Y_{i,i+1}=\;J_{i,i+1}\sin\theta ,\;\mathrm{for}\;i=1\;\mathrm{to}\;n-1,\;Y_{t}=Y_{1\;}-\frac{Y_{1,2\;}^{2}}{Y_{1}}\;\mathrm{and}\;\theta_{t}=\frac{\sin^{-1}(\sqrt{\frac{Y\sin^{2}\theta }{Y_{0}g_{0}g_{1}}})}{1-\frac{FBW}{2}}$$

In this paper, *n* is chosen to be 3, and since the filter is symmetrical, the three resonators have the same *Y*_1_. The *g*’s values of Chebyshev response lowpass prototype with 0.1 dB passband ripple are *g*_0_ = *g*_4_ = 1, *g*_1_ = *g*_3_ = 1.0316, *g*_2_ = 1.1474. To compensate for the frequency shift of the tapping effect at the input and output, a capacitance *C_t_* is loaded at the input and output resonator, and it is equal to
(6)$$C_{t}=\frac{cos\theta_{t}\sin^{3}\theta_{t}}{w_{0}\;Y_{t}\left(\frac{1}{Y_{0}^{2}}+\frac{\cos^{2}\theta_{t}\sin^{2}\theta_{t}}{Y_{t}}\right)}$$

Due to this capacitance, a small length is added to the input and output resonators and can be calculated using
(7)$$\;\Delta L_{C}=\frac{\lambda_{g}}{2\pi }\tan^{-1}(\frac{2\pi F_{c}C_{t}}{Y_{1}}\;)$$
where λ_g_ is the guided wavelength at the center frequency, *F_C_*, of the required band. The coupling factor (*k_i_*,*i*+1) is used to determine the space between two adjacent resonators (*S*) and can be found using
(8)$$k_{i,i+1}=\frac{Z_{0ei,i+1\;}-Z_{0oi,i+1\;}}{Z_{0ei,i+1\;}+Z_{0oi,i+1\;}}$$
where $Z_{0e1,2\;}=\frac{1}{Y_{1}-Y_{1.2}}$, $Z_{0O1,2\;}=\frac{1}{Y_{1}+Y_{1.2}}$, $Z_{0ei,i+1\;}=\frac{1}{2Y_{1}-\frac{1}{Z_{0ei-1,i\;}}-Y_{i.i+1}-Y_{i-1.i}}$, for *i* = 2 to *n* − 2 $Z_{0ei,i+1\;}=\frac{1}{{\;}_{2Y_{i,\;i+1}+\frac{1}{Z_{0ei,i+1\;}}}\;}$, for *i* = 2 to *n* − 2, $Z_{0en-1,n\;}=\frac{1}{Y_{1}-Y_{n-1.n}},\;Z_{0on-1,n\;}=\frac{1}{Y_{1}+Y_{n-1.n}}\;$.

It should be mentioned here, in full-wave electromagnetic simulations such as HFSS, the relation between *k*_1,2_ = *k*_2,3_ and *S* can be extracted using [1]
(9)$$\;k_{12}=k_{23}=\frac{w_{0}}{\Delta w\pm 90{}^{\circ}}\;$$
where $w_{0}$ and Δw±90 are the angular center frequency and absolute bandwidth between ±90° points, respectively.

Based on the design Equations at (5)–(9), at *f_c_* = 3.95 GHz and 6.55 GHz with fractional bandwidth (FBW) = 12.66%, and 17.56%, the third order UTL IBPF at (3.7–4.2 GHz) and (5.975–7.125 GHz), respectively, is firstly designed, then NTLs theory is applied for compactness. For both filters, the relation between the coupling factor, *k*_12_
*= k*_23_ and *S*, is shown in Figure 4. Different parametric studies are performed using HFSS on UTL IBPF at (3.7–4.2 GHz) and (5.975–7.125 GHz) as shown in Figure 5 and Figure 6, respectively. The theoretical and optimized parameters are demonstrated in Table 1, where *L*_*res*1_, *L*_*res*2_, *L_t_*, and *W_p_* are the length of the first and second resonators, tapping length, and width of the ports, respectively. As noticed from Figure 5a, the band shifts toward low (*Fl*) or high (*Fh*) frequencies as *L_res_* increases or decreases, respectively, and the impedance matching level corresponding to the optimized value of *L*_*res*2_ = 10.1 mm (where S_11_ = S_22_ < −11.84 dB at 3.57–4.24 GHz) is better than that corresponding to the theoretical value *L*_*res*2_ = 11.38 mm (where S_11_= S_22_ < −9.19 dB at 3.16–3.92 GHz). The return loss is better than 10 dB when *W_res_* > 1.84 mm as shown in Figure 5b. Although the impedance matching at *W_res_* = 2.06 mm (where S_11_ = S_22_ < −11.92 dB at 3.54–4.2 = 0.658 GHz ) is 0.67% better, the BW is wider at *W_res_* = 1.95 mm. Figure 5c illustrates that to obtain the BW enhancement, the coupling between resonators should be increased by decreasing *S* as at *S* = 0.8 mm (where S_11_= S_22_ < −11.84 dB at 3.57–4.24 GHz), in which the BW is wider than at the theoretical *S* = 0.95 mm (where S_11_ = S_22_ < −11.84 dB at 3.63–3.92 GHz) despite its better impedance matching. Although the BW at *L_t_* = 2.3 mm (where S_11_ = S_22_ < −9.1 dB at 3.52–4.39 GHz) is 23% wider, *L_t_* = 2.75 mm is selected due to its 23.14% better impedance matching as shown in Figure 5d. When the radius of via hole, *rvia* = 0.425 mm, the return loss is better than 10 dB within the required band as indicated in Figure 5e.

Figure 6a shows that S_11_ at the theoretical *L*_*res*2_ is <−10 dB, but the BW shifts toward *Fl* or *Fh* when it is greater or less than 6.81 mm, respectively. In Figure 6b at *W_res_* = 2.25 mm, the higher frequency of the band (7.125 GHz) is not achieved. Although the impedance matching at *W_res_* = 2.35 mm is better than that at *W_res_* = 2.3 mm, 7.125 GHz is covered at *W_res_* = 2.3 mm. Enhanced BW is obtained at *S* = 0.4 mm (where S_11_ = S_22_ < −10.7 dB at 5.28–7.17 GHz) as depicted in Figure 6c, which is better than that at the theoretical *S* = 0.55 mm (where S_11_ = S_22_ < −11 dB at 6.59–6.44 GHz) in terms of BW. Figure 6c illustrates that at *L_t_* =1.9 mm, there is a mismatch (where S_11_ = S_22_ < −7.62 dB at 5.19–7.41 GHz) and the best impedance matching is obtained at *L_t_* = 2.4 mm. Moreover, it can be noticed that when *L_t_* > 2.4 mm, the obtained BW is less than the required one. An acceptable impedance matching is obtained at *W_p_* = 1.78 mm as shown in Figure 6d. Finally, from Figure 6e, *rvia* is selected to be 0.75 mm for its better BW.

The NTLs concept is then applied to UTL IBPFs in which each λ/4 UTL resonator is replaced with its equivalent NTL resonator. Size reductions of 16.88% and 16.83% are achieved in the first (symmetrical to the third resonator) and second λ/4 resonators of 3.95 GHz IBPF, respectively. However, 16.46% and 16.33% size reductions are achieved in the first (symmetrical to the third resonator) and second λ/4 resonators of 6.55 GHz UTL IBPFs, respectively. The optimized *C*_n_s values at both frequency bands are shown in Table 2. Further parametric studies are performed on the designed 3.95 GHz and 6.55 GHz NTL IBPFs as shown in Figure 7 and Figure 8, respectively. *W*_*p*1_ and *W*_*p*2_ in Figure 8 are the widths of the input and output ports, respectively.

The 2D configuration and prototype of the proposed 3.95 GHz and 6.55 GHz UTL and NTL IBPFs are shown in Figure 9 and Figure 10, respectively. It should be mentioned here that the difference between the resulting resonators of NTL IBPF is due to their different optimized *C_n_*s as illustrated in Table 2. The circuit areas for the proposed 3.95 GHz and 6.55 GHz UTL and NTL IBPF are 18.8 × 12.2 mm^2^ (0.41 λg × 0.27 λg) and 14.7 × 9.58 mm^2^ (0.32 λg × 0.21 λg), 16.7 × 9.83 mm^2^ (0.61 λg × 0.36 λg), and 13.9 × 7.02 mm^2^ (0.51 λg × 0.26 λg), in which 36.6% and 40.53% size reductions are obtained, respectively.

## 4. Result and Discussion

The comparison between the simulated and measured results of the UTL and NTL filters is illustrated in Figure 11 and Table 3. The proposed 3.95 GHz and 6.55 GHz NTL IBPFs provide enhanced BWs of 0.05 GHz and 0.58 GHz and second harmonics suppression (HS) up to 11.25 GHz and 20 GHz, respectively. The discrepancy between the simulated and measured results is due to fabrication tolerance, imperfect soldering of SMA connectors, imperfect filling of via holes, and the difference between the simulation and real measurement environments. It should be mentioned here that due to the nonuniformity of the NTL resonators and the small space between them (*S*), they will be overlapped using chemical etching. This overlapping is removed manually and resulted in *S* being different than the optimized one, which can be noticed in the discrepancy between the simulated and measured results of the NTL filters. The acceptable response of the NTL IBPF is obtained thanks to the efficient use of the MATLAB built-in function “fmincon.m”, which optimizes the *Z(z )’s Cns* of the NTL resonators by reducing the error function in (4). In addition to the 36.6% and the 40.53% total area reductions achieved using NTL theory, they provide good performance, especially in suppressing harmonics as compared to the NTL HPBFs proposed in [9,10]. Due to the different lengths between the UTL and NTL resonators at both bands, there is only a slight phase shift in the filter’s response, as shown in Figure 11c,f; this implies the effectiveness of applying the NTL theory in reducing the size of the NTL IBPF, which makes it a good candidate for CRN applications.

A comparison with other works in the literature at different frequency bands is shown in Table 4. As compared to the compactness techniques used in other references, and in order to avoid any fabrication difficulties or high-cost substrate materials, NTL theory is simply applied for the first time to compact the IBPF’s size providing better compactness than [30], a better S_11_ response and compactness than [18] and a better S_12_ response and wider FBW than [27,31]. Moreover, it provides wider FBW, and a better S_12_ response and compactness than [8,18,25,26]. The IBPF in [28] outperforms the proposed one due to the wet etching fabrication process with a film for negative development. This method can be used in our future work along with the NTL SIR.

## 5. Conclusions

At two 5G low-frequency bands (3.7–4.2 GHz and 5.975–7.125 GHz), a compact nonuniform transmission line (NTL) interdigital bandpass filter (IBPF) appropriate for modern wireless communication systems is analyzed, designed, and tested in this paper. At the first and second frequency bands, 36.6% and 40.53% total circuit area reductions, an acceptable level of impedance matching and up to 11.25 GHz and 20 GHz second harmonics suppressions, are achieved, respectively. The proposed filter is a part of our future reconfigurable 5G narrow bands and UWB antenna project. Future work may concentrate on the size reduction of other types of BPFs such as combline filters or other microwave components at 5G high-frequency bands using NTLs theory.

## Figures and Tables

**Figure 1 micromachines-13-02013-f001:**
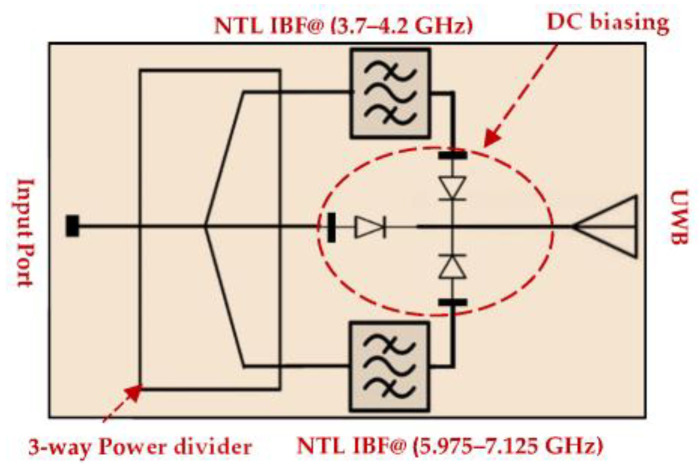
Future project: compact reconfigurable 5G and UWB antenna.

**Figure 2 micromachines-13-02013-f002:**
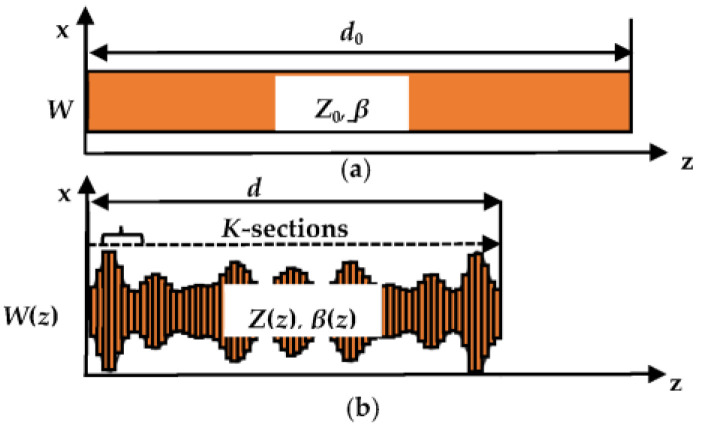
(**a**) UTL and (**b**) NTL with its subdivision into *K* uniform sections.

**Figure 3 micromachines-13-02013-f003:**
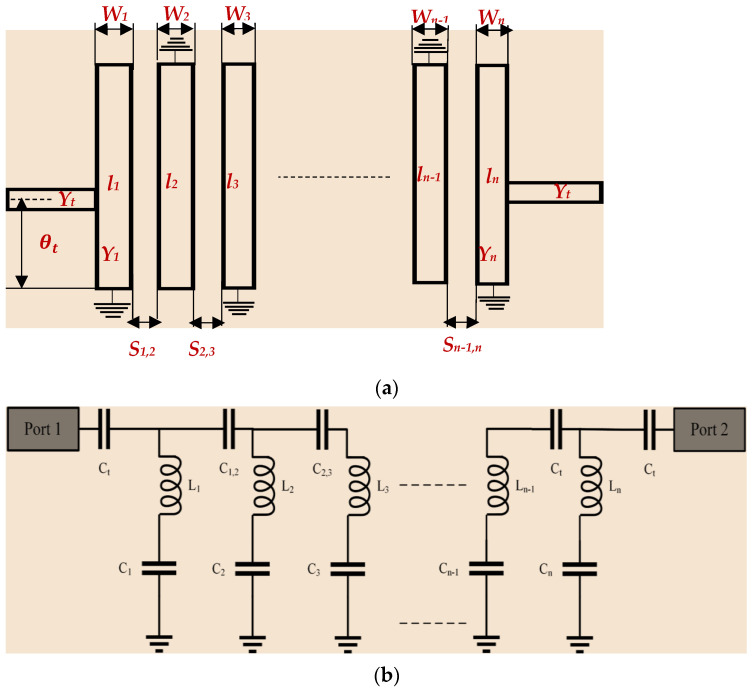
*n* order UTL IBPF (**a**) layoutand (**b**) equivalent circuit.

**Figure 4 micromachines-13-02013-f004:**
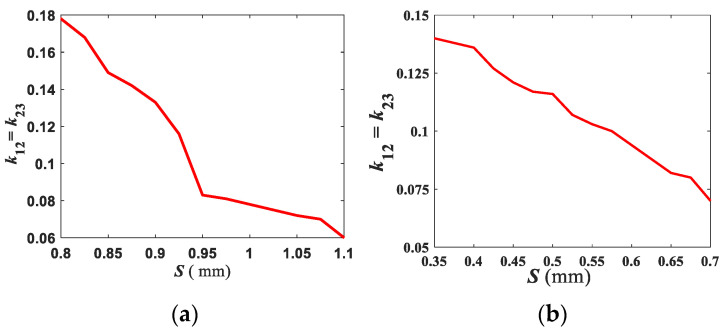
Variation of mutual coupling due to the space between adjacent resonators of UTL IBPF at (**a**) (3.7–4.2 GHz) and (**b**) (5.975–7.125 GHz).

**Figure 5 micromachines-13-02013-f005:**
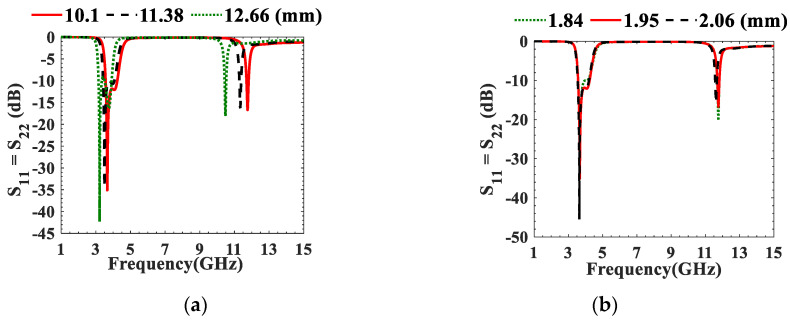
Parametric studies of the proposed UTL IBPF at (3.7–4.2 GHz) on (**a**) *L*_*res*2_, (**b**) *W_res_*, (**c**) *S*, (**d**) *L_t_*, and (**e**) *rVia*.

**Figure 6 micromachines-13-02013-f006:**
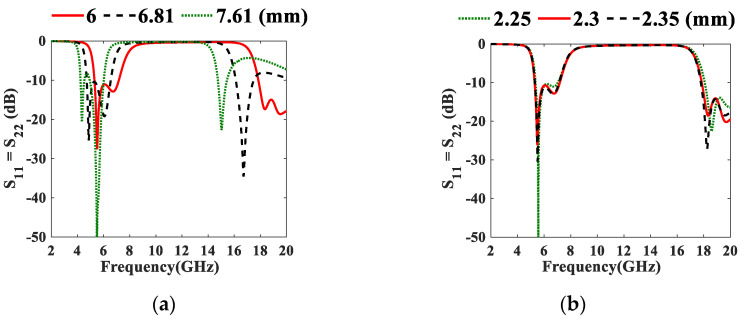
Parametric studies of the proposed 6.55 GHz UTL IBF on (**a**) *L*_*res*2_, (**b**) *W_res_*, (**c**) *S*, (**d**) *L_t_*, (**e**) *W_p_* and (**f**) *rVia*.

**Figure 7 micromachines-13-02013-f007:**
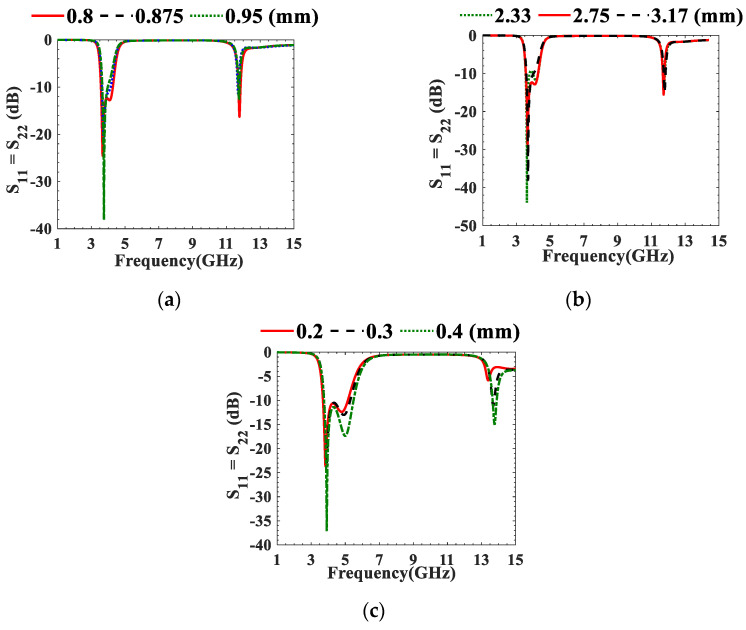
Parametric studies of the proposed NTL IBPF at (3.7–4.2 GHz) on (**a**) *S*, (**b**) *L_t_*, and (**c**) *rvia*.

**Figure 8 micromachines-13-02013-f008:**
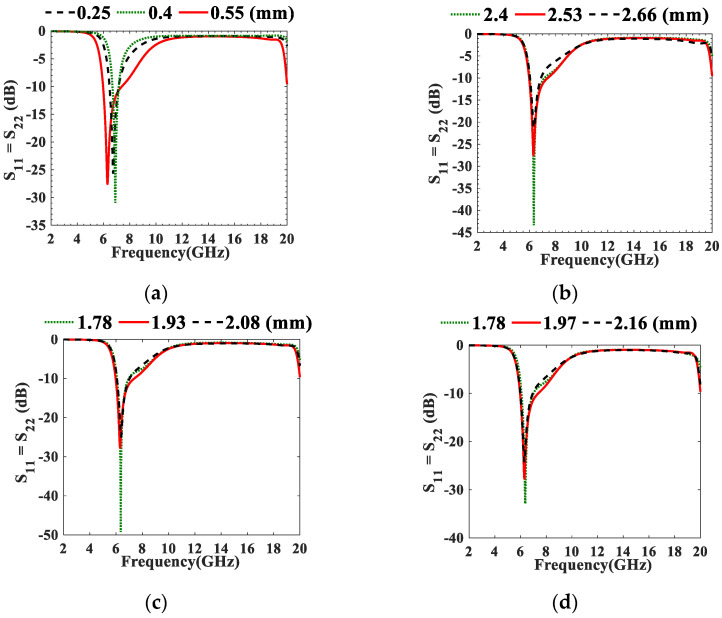
Parametric studies of the proposed NTL IBPF at (5.975–7.125 GHz) (**a**) *S*, (**b**) *L_t_*, (**c**) *W*_*p*1_, (**d**) *W*_*p*2_, and (**e**) *rvia*.

**Figure 9 micromachines-13-02013-f009:**
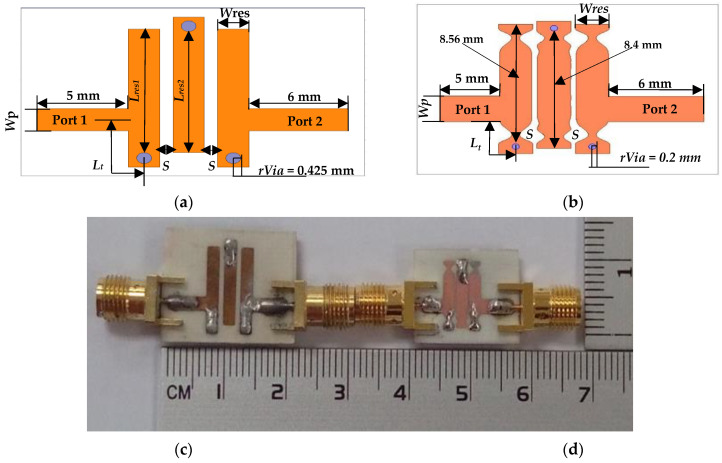
(**a**,**b**) 2D configuration and (**c**,**d**) photograph of the proposed (3.7–4.2 GHz) UTL and NTL IBPFs, respectively.

**Figure 10 micromachines-13-02013-f010:**
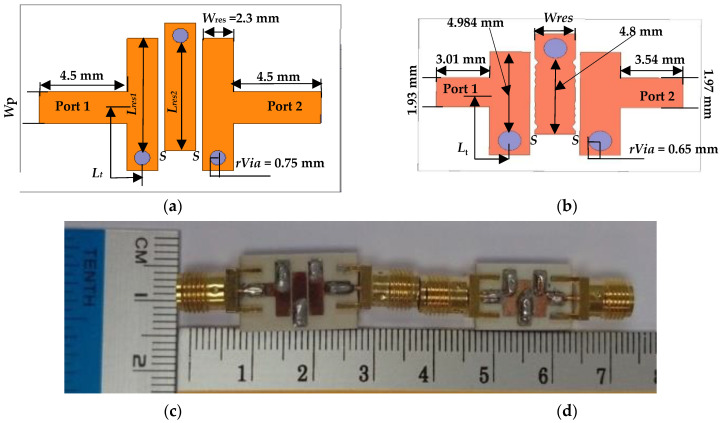
(**a**,**b**) 2D configuration and (**c**,**d**) photograph of the proposed (5.975–7.125 GHz) UTL and NTL IBPFs, respectively.

**Figure 11 micromachines-13-02013-f011:**
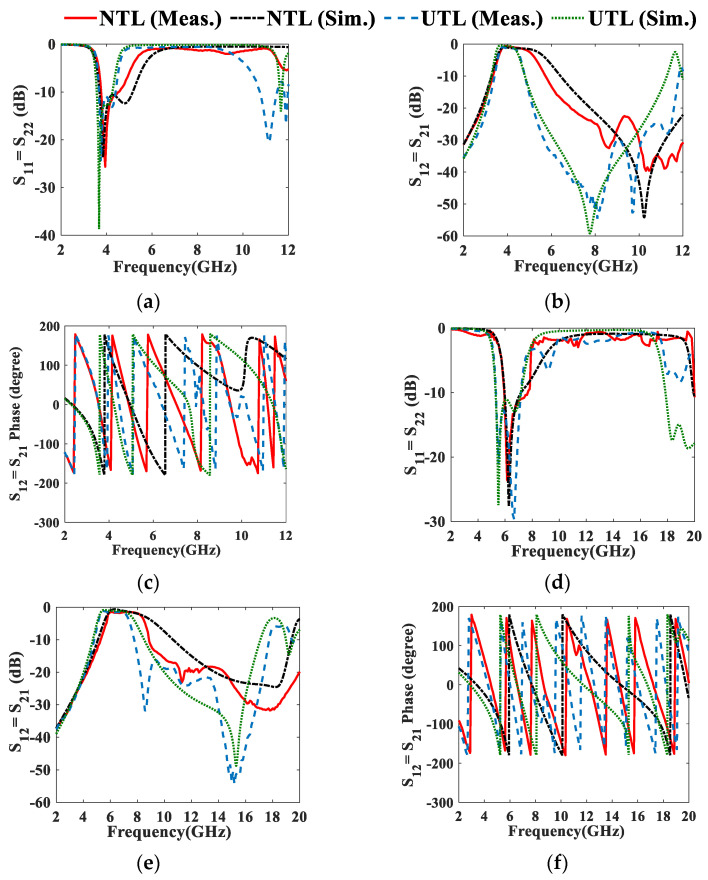
Simulated and measured (**a**,**b**) S_11_, (**c**,**d**) S_12,_ and (**e**,**f**) phase of the proposed (3.7–4.2 GHz) and (5.975–7.125 GHz) NTL and UTL IBPFs.

**Table 1 micromachines-13-02013-t001:** Theoretical and optimized parameters of 3.95 GHz and 6.55 GHz UTL IBPFs.

Parameters	3.95 GHz UTLIBF		6.55 GHz UTLIBF	
	Calculated	Optimized	Calculated	Optimized
*k* _12_ *= k* _23_	0.091	-	0.127	-
*L*_*res*1_ (mm)	11.58	10.3	7.0382	6.23
*L*_*res*2_ (mm)	11.38	10.1	6.81	6
*W_res_* (mm)	1.843	1.95	2.245	2.3
*S* (mm)	0.95	0.8	0.55	0.4
*L_t_* (mm)	2.332	2.75	1.9	2.4
*rVia* (mm)	-	0.425	-	0.75
*L*_*p*1_ (mm)	-	3.5	-	3
*L*_*p*2_ (mm)	-	5.5		3.5
*W_p_* (mm)	1.819	1.819	1.819	1.78

**Table 2 micromachines-13-02013-t002:** Optimized Fourier coefficients for λ/4 NTL IBPF’s resonators.

at (3.7–4.2 GHz)						
$\mathrm{Constraints}:\;1\;\leq \;\bar{Z}\left(z\right)\;\leq \;2.393$						
C_0_		C_1_	C_2_	C_3_	C_4_	C_5_
1st	0.2896	0.3420	−0.0105	−0.1674	−0.1339	−0.0670
2nd	0.0602	0.0825	0.0496	0.0130	−0.0038	−0.0188
C_6_		C_7_	C_8_	C_9_	C_10_	
1st	−0.0299	−0.0282	−0.0636	−0.0848	−0.0462	
2nd	−0.0335	−0.0514	−0.0493	−0.0342	−0.0143	
**at (5.975–7.125 GHz)**						
$\mathrm{Constraints}:\;1\;\leq \;\bar{Z}\left(z\right)\;\leq \;2.711$						
C_0_		C_1_	C_2_	C_3_	C_4_	C_5_
1st	0.0043	0.0007	−0.0002	0.0004	0.0004	−0.0003
2nd	0.0222	0.0180	−0.0028	0.0026	0.0018	−0.0188
C_6_		C_7_	C_8_	C_9_	C_10_	
1st	−0.0004	−0.0001	−0.0003	−0.0009	−0.0011	
2nd	−0.0054	−0.0016	−0.0056	−0.0140	−0.0095	

**Table 3 micromachines-13-02013-t003:** Comparison between the simulated and measured results for the designed UTL and NTL IBFs.

Parameters	Sim.	Meas.	Sim.	Meas.
at (3.7–4.2 GHz)			at (5.975–7.125 GHz)	
S_11_ = S_22_ (NTL)	<−10.63 dB.at 3.7–5.15 GHz, H.S up to 11.8 GHz	<−11.2 dB.at 3.7–4.25 GHz, H.S up to 11.25 GHz	<−10.45 dB.at 5.93–7.39 GHz H.S up to 18 GHz	<−11.27 dB at 5.94–7.67 GHz, H.S up to 20 GHz
S_11_ = S_22_ (UTL)	<−11.84 dB at 3.57–4.24 GHz, H.S up to 11.8 GHz	<−10.63 dB at 3.66–4.4 GHz, H.S up to 11.12 GHz	<−11.05dB at 5.3–7.164 GHz, H.S up to 18 GHz	<−13.91 dB at 5.34–7.163 GHz, H.S up to 20 GHz
S_12_ = S_21_ (NTL)	−1.34 dB at *F_C_* = 4.43 GHz	−0.86 dB at *F_C_* = 3.98 GHz	−0.8 dB at *F_C_* = 6.68 GHz	−1.7 dB at *F_C_* = 6.81 GHz
S_12_ = S_21_ (UTL)	−0.64 dB at *F_C_* = 3.91 GHz	−1.26 dB at *F_C_* = 4.03 GHz	−1.09 dB at *F_C_* = 6.25 GHz	−1.78 dB at *F_C_* = 6.25 GHz

**Table 4 micromachines-13-02013-t004:** A comparison to other works in the literature at different frequency bands.

Ref., Layers Order	h (mm)/εr	Technique	3dB FBW, Frequency Band (GHz)	S_11_ (dB)	S_12_ (dB)	H.S Up to GHz	Area mm^2^, λg^2^
This work 3rd	0.813/3.55	NTLs	13.82%, 3.7–4.25	−10.52	−0.86	11.25	16.7 × 9.83, 0.32 × 0.21
			25.40%, 5.94–7.67	−11.27	−1.7	18	13.9 × 7.02, 0.51 × 0.26
[31], 1, 3rd	IPDs	TQVs	9.42%, 27.2–29.89	−15.17	−1.66	74	1.715 × 7.6, 0.15 × 0.15
[30], 1, 7th	IPDs	TSV	58.89%, 61.97–113.7	−20	−1.3	87	0.5 × 0.34, 0.48 × 0.33
[28], 1, 3rd	0.54/2.54	spiral and folded SIR	180%, 0.2–1.42	−17.1	−0.043	NA	13.8 × 5.98, 0.05 × 0.02
[16],2, 6th	0.40/11.9	MEMS technology	7.8%, 19.6–21.2	−15	−1.98	NA	7 × 3, 1.34 × 0.57
[25], 4, 2nd	0.0034/2.9	Inject printing on LCPT	17%, 10.52–12.48	−12	−2.2	NA	5.28 × 3.32, 0.32 × 0.2
[8], 1, 4th	0.381/9.8	High εr substrate	15%, 7.4–8.6	−20	−1.5	NA	5 × 5, 0.34 × 0.34
[26], 1, 5th	0.508/3.66	λ/4 and λ/2 resonators	4.5%, 9.85–10.3	−14.68	−4.56	NA	28.30 × 10.63, 1.51 × 0.60
[18],1, 5th	1.27/10.2	Spurlines	64%, 0.99–1.96	−10.93	−0.83	8	15.5 × 22, 0.24 × 0.35
[27], 1, 2nd	0.05/3	Ultra-thin LCP	8.19%, 9–9.77	−22	−2.3	NA	5.4 × 3.2, 0.16 × 0.26

NA: not available.

## Data Availability

Not applicable.

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
