# Peer review of "Compact 5G Nonuniform Transmission Line Interdigital Bandpass Filter for 5G/UWB Reconfigurable Antenna"

_micromachines, 2022, doi:10.3390/mi13112013_

Round 1

Author Response

Thank you very much for your comments and suggestions

Attached is the response

Reviewer 2 Report

In this article, a compact non-uniform Transmission Line (NTL) Interdigital Band Pass Filter (IBF) at two 5G low-frequency bands (3.7 – 4.2 GHz and 5.975 – 7.125 GHz) is analyzed, designed, and tested for a modern wireless communication system. The manuscript has the potential to be accepted for publication. However, updates are required before the resubmission.

1.      The authors should add the equivalent circuit model of the proposed bandpass filter with simulation results in the manuscript to show the working principle and effectiveness of the proposed bandpass filter.  

2.      The value of an important parameter of the antenna design (tanδ of the substrate) is missing in the manuscript. Please provide the value of tanδ of the substrate in the manuscript.

3.      The manuscript represents a reconfigurable antenna for the UWB and lower 5G band applications. However, there is a lack of literature review about the UWB and lower 5G antennas and especially the benefit of these frequency bands. Here are some suggestions about the UWB and lower 5G band ranges and their usage, respectively. Isolation and Gain Improvement of a Rectangular Notch UWB-MIMO Antenna, Sensors, 2022; and Isolation enhancement of a metasurface-based MIMO antenna using slots and shorting pins, IEEE Access, 2021. The authors also can find related articles by internet search.

4.      Add more recently published similar work in the comparison table (Table 4)

5.      Please take care of the formatting and typos. (a) All data in Table 1 should be on the same page, the last row is presented on the next page, (b) Figure data and figure title of Figure 5 should be on the same page, (c)Table caption of Table 2 should be on the top of the table instead of the below of the table, the same comment for the Table 4, (d) The references are not formatted accordingly to the journal template. Update them accordingly.

Author Response

All the required changes are highlighted in the manuscript in yellow color.

Reviewer 1

Response to the comments

Thank you very much for your effort in reviewing our manuscript. The followings are our response to the given comments

Comments and Suggestions for Authors

In this article, a compact non-uniform Transmission Line (NTL) Interdigital Band Pass Filter (IBF) at two 5G low-frequency bands (3.7 – 4.2 GHz and 5.975 – 7.125 GHz) is analyzed, designed, and tested for a modern wireless communication system. The manuscript has the potential to be accepted for publication. However, updates are required before the resubmission.

  1. The authors should add the equivalent circuit model of the proposed bandpass filter with simulation results in the manuscript to show the working principle and effectiveness of the proposed bandpass filter.  

Reply

The equivalent circuit for IBF is added as part of Figure 3 (Figure 3(b)) to Section 1, Second paragraph, page 5, lines 217-231

  1. The value of an important parameter of the antenna design (tanδ of the substrate) is missing in the manuscript. Please provide the value of tanδ of the substrate in the manuscript.

Reply

The required loss tangent value tanδ is added to Section 1, Second paragraph, page 2, line 103 as follows

The chosen substrate material is Rogers RO4003C (Ɛ? = 3.55, tanδ= 0.0027 and h = 0.813 mm)

  1. The manuscript represents a reconfigurable antenna for the UWB and lower 5G band applications. However, there is a lack of literature review about the UWB and lower 5G antennas and especially the benefit of these frequency bands. Here are some suggestions about the UWB and lower 5G band ranges and their usage, respectively. Isolation and Gain Improvement of a Rectangular Notch UWB-MIMO Antenna, Sensors, 2022; and Isolation enhancement of a metasurface-based MIMO antenna using slots and shorting pins, IEEE Access, 2021. The authors also can find related articles by internet search.

Reply

The whole manuscript is about designing the compact (5G low frequency bands) 3.95 GHz and 6.55 GHz IBFs using the NTL theory for the first time. Then because of its good properties in terms of size, wide bandwidth and high order mode harmonic suppressions, the aim for future work is to integrate it with UWB antenna to get a reconfigurable UWB/5G antenna, where there is enough space to add the DC biasing circuits. For this purpose, the main literature is about IBF at different frequency bands. Although the required references are out of the scope of our desired future work, they are included with other recent references as an application for UWB and 5G low-frequency band technologies.  To spot some light on reconfigurable UWB/narrow band antennas, three recent references are added. The required references and the reconfigurable references are added to the manuscript as follows

Due to the importance of UWB technology [1–8]and 5G low-frequency bands [9–13] in the modern wireless communication system in terms of high data rate, compatibility with consumer demand, and low latency, switching between these bands is more practical in terms of size and time. For this purpose, as shown in Figure 1, the designed filter will be utilized as a part of our future project that aims to have a compact reconfigurable UWB/5G low-frequency band suitable for cognitive radio networks (CRNs) application. Some recent using this concept can be found in [14–18].

This required information is added to Section 1, Second paragraph, pages 2-3, lines 80-100

  1. Add more recently published similar work in the comparison table (Table 4)

 Reply

The table already contains a recent reference from 2021, however, other recent related two refences are added to the introduction and Table 4 as follows

Introduction:

Recently, millimeter-wave (mmW) IBFs based on integrated passive devices (IPDs) are designed in [19,20]. In [19], through-quartz vias (TQVs) are added to the proposed IBF in[20] for coupling adjustment and size reduction. However, for IBF’s compactness, wideband, and better harmonics suppressions, through-silicon via (TSV)-based 3-D integrated circuit (3-D IC) technology is used in [19].

This required information is added to Section 1, Second paragraph, pages 2, lines 69-73

Ref., layers

order

h(mm)/ εr

Technique

3dB  FBW, Frequency Band (GHz)

S11 (dB)<

S12 (dB)

H.S Up to GHz

area

mm2, λg2

[20], 1, 3rd

IPDs

TQVs

9.42 %,  27.2 – 29.89

-15.17

-1.66

74

1.715 × 7.6, 0.15 × 0.15

[19], 1, 7th

IPDs

TSV

58.89 %,  61.97 –113.7

-20

-1.3

87

0.5 × 0.34, 0.48 × 0.33

Table 4: 4th and 5th rows

  1. Please take care of the formatting and typos. (a) All data in Table 1 should be on the same page, the last row is presented on the next page, (b) Figure data and figure title of Figure 5 should be on the same page, (c)Table caption of Table 2 should be on the top of the table instead of the below of the table, the same comment for the Table 4, (d) The references are not formatted accordingly to the journal template. Update them accordingly.

Reply

(a) Table 1 position is adjusted so all the rows in the same page

(b) Figure 5 position is adjusted so all the related data on one page

(c) The caption for Table 2 and 4 are moved to the top

(d)The reference format is modified as required

References

  1. Saleh, S.; Ismail, W.; Sorfina, I.; Abidin, Z.; Jamaluddin, H.; Bataineh, M.H.; Al-zoubi, A.S. Novel Compact UWB Vivaldi Nonuniform Slot Antenna With Enhanced Bandwidth. 2022, 70, 6592–6603.
  2. Saleh, S.; Ismail, W.; Abidin, I.S.Z.; Jamaluddin, M.H.; Bataineh, M.H.; Alzoubi, A.S. Compact UWB Vivaldi Tapered Slot Antenna. Alexandria Engineering Journal 2022, 61, 4977–4994, doi:10.1016/j.aej.2021.09.055.
  3. Abbas, A.; Hussain, N.; Sufian, M.A.; Jung, J.; Park, S.M.; Kim, N. Isolation and Gain Improvement of a Rectangular Notch Uwb-Mimo Antenna. Sensors 2022, 22, doi:10.3390/s22041460.
  4. Shabbir, T.; Saleem, R.; Al-Bawri, S.S.; Shafique, M.F.; Islam, M.T. Eight-Port Metamaterial Loaded UWB-MIMO Antenna System for 3D System-in-Package Applications. IEEE Access 2020, 8, 106982–106992.
  5. Kissi, C.; Särestöniemi, M.; Kumpuniemi, T.; Myllymäki, S.; Sonkki, M.; Mäkelä, J.-P.; Srifi, M.N.; Jantunen, H.; Pomalaza-Raez, C. Receiving UWB Antenna for Wireless Capsule Endoscopy Communications. Progress In Electromagnetics Research C 2020, 101, 53–69.
  6. Balderas, L.I.; Reyna, A.; Panduro, M.A.; del Rio, C.; Gutierrez, A.R. Low-Profile Conformal UWB Antenna for UAV Applications. IEEE Access 2019, 7, 127486–127494, doi:10.1109/ACCESS.2019.2939511.
  7. Parameswari, S.; Chitra, C. Compact Textile UWB Antenna with Hexagonal for Biomedical Communication. J Ambient Intell Humaniz Comput 2021, 1–8.
  8. Kanagasabai, M.; Sambandam, P.; Alsath, M.G.N.; Palaniswamy, S.; Ravichandran, A.; Girinathan, C. Miniaturized Circularly Polarized UWB Antenna for Body Centric Communication. IEEE Trans Antennas Propag 2021.
  9. Sufian, M.A.; Hussain, N.; Askari, H.; Park, S.G.; Shin, K.S.; Kim, N. Isolation Enhancement of a Metasurface-Based MIMO Antenna Using Slots and Shorting Pins. IEEE Access 2021, 9, 73533–73543, doi:10.1109/ACCESS.2021.3079965.
  10. Saleh, S.; Ismail, W.; Zainal Abidin, I.S.; Jamaluddin, M.H. Compact 5G Hairpin Bandpass Filter Using Non-Uniform Transmission Lines Theory. Appl Comput Electromagn Soc J 2021, 36, doi:10.47037/2020.ACES.J.360202.
  11. Feng, Y.; Li, J.Y.; Zhang, L.K.; Yu, X.J.; Qi, Y.X.; Li, D.; Zhou, S.G. A Broadband Wide-Angle Scanning Linear Array Antenna With Suppressed Mutual Coupling for 5G Sub-6G Applications. IEEE Antennas Wirel Propag Lett 2022, 21, 366–370, doi:10.1109/LAWP.2021.3131806.
  12. Tang, X.; Chen, H.; Yu, B.; Che, W.; Xue, Q. Bandwidth Enhancement of a Compact Dual-Polarized Antenna for Sub-6G 5G CPE. IEEE Antennas Wirel Propag Lett 2022, 21, 2015–2019, doi:10.1109/LAWP.2022.3188751.
  13. Wang, S.; Hao, H.; Ma, X.; Cheng, H.; Huang, X. Wideband Circularly Polarized Array Antenna Based on Sequential Phase Feeding Metasurfaces for 5G (Sub-6G) Applications. J Electromagn Waves Appl 2022, doi:10.1080/09205071.2022.2110948.
  14. Saleh, S.; Ismail, W.; Abidin, I.S.Z.; Jamaluddin, M.H.; Bataineh, M.; Alzoubi, A. Compact Reconfigurable Ultra Wide Band and 5G Narrow Band Vivaldi Tapered Slot Antenna. In Proceedings of the 2020 IEEE International RF and Microwave Conference, RFM 2020 - Proceeding; Kuala Lumpur,Malaysia, 2020.
  15. Potti, D.S.; Balaji, P.; Gulam Nabi Alsath, M.; Savarimuthu, K.; Selvam, U.; Valavan, N. Reconfigurable Bow Tie-Based Filtering Antenna for Cognitive Radio Applications. International Journal of RF and Microwave Computer-Aided Engineering 2020, 30, 1–10, doi:10.1002/mmce.22208.
  16. Niture, D. v.; Mahajan, S.P. A Compact Reconfigurable Antenna for UWB and Cognitive Radio Applications. Wirel Pers Commun 2022, 125, 3661–3679, doi:10.1007/s11277-022-09729-4.
  17. Deng, J.; Hou, S.; Zhao, L.; Guo, L. A Reconfigurable Filtering Antenna with Integrated Bandpass Filters for UWB/WLAN Applications. IEEE Trans Antennas Propag 2017, 66, 401–404.
  18. Kantemur, A.; Abdelrahman, A.H.; Xin, H. A Novel Compact Reconfigurable UWB Antenna for Cognitive Radio Applications. 2017 IEEE Antennas and Propagation Society International Symposium, Proceedings 2017, 2017-Janua, 1369–1370, doi:10.1109/APUSNCURSINRSM.2017.8072727.
  19. Wang, F.; Zhang, K.; Yin, X.; Yu, N.; Yang, Y. A Miniaturized Wideband Interdigital Bandpass Filter With High Out-Band Suppression Based on TSV Technology for W-Band Application. IEEE Trans Very Large Scale Integr VLSI Syst 2022.
  20. Liu, N.; Liu, X.; Liu, Y.; Yang, Y.; Zhu, Z. Compact Interdigital Bandpass Filter, Diplexer, and Triplexer Based on Through Quartz Vias (TQVs). IEEE Trans Compon Packaging Manuf Technol 2022, 12, 988–997, doi:10.1109/TCPMT.2022.3178411.

Reviewer 3 Report

Overall, a good paper, however, minor revisions are necessary.  See the attached file.

Author Response

Thank you very much for your valuable comments, all the required changes are done as highlighted throughout the manuscript. A point-by-point response can be found as replies to your pdf

Round 2

Reviewer 1 Report

Compact 5G Nonuniform Transmission Lines Interdigital Band Pass Filter for 5G/UWB reconfigurable antenna

Review report – 2nd round 

Abstract:

Please note that they are two prototypes. “In this study at two different fifth generation (5G) low-frequency bands (3.7– 4.2 GHz and 13 5.975–7.125 GHz) and based on non-uniform transmission lines (NTLs) theory, third order compact interdigital band pass filter (IBPF) is analyzed, designed, and fabricated. The compact proposed filter is considered as a good candidate

Author should mention at the beginning that these two prototypes are composed of three quarter-wave resonators.

Please write “less” not “<” in: “S11 is appeared to be < -10.53 dB and < -11.27 dB through 3.7–4.25 GHz…”

Section I (Introduction):

Author has already mentioned this in Abstract so he can remove it: “In this study, HFSS is used for the simulation

Section III

Please, simulated parametric study is performed over a frequency interval not on specified f0 frequency.  Different parametric studies are performed using HFSS on 3.95 GHz and 6.55 GHz…”

In figures 5 and 6, Author has to put the geometrical parameter’ symbol on the graph representing S11 and S22 variations corresponding to each parameter cited in the caption of figures 5 and 6, because there are many graphs (5 and 6 graphs respectively) representing S-parameters variations in term of frequencies.

Section IV

Please revise this sentence: “…and this besides the previous factors explain the obvious difference between the simulated and measured BW of NTL filters.”

This has to be revised please: “As compared to the compactness techniques used in other references and  avoiding any fabrication difficulties or high-cost substrate materials, NTL theory is simply applied for the first time….” Suggestion: : “As compared to the compactness techniques used in other references and  in order to avoid any fabrication difficulties or high-cost substrate materials, NTL theory is simply applied for the first time….”

Section VI (Conclusion)

Author did not reply on our following comment: But if we see the measured results of figure 11, concerning NTL method (solid red color), I think S11 is more inferior than -10.53 and -11.27 dB, respectively at 3.95 GHz and 6.55GHz as shown by the following figures (a) and (d) of figure 11. So can author clarify this?

(see figures in attached PDF file)

Author Response

Thanks for your comments.

Attached is the required response

Reviewer 2 Report

All the comments are addressed properly in this version. The revised version manuscript can be accepted for publication after the following minor edit.

The authors should redraw Figure 3 (equivalent circuit model) by using any professional software (Microsoft Visio Drawing / AutoCAD etc.). The schematic in figure 3 doesn’t show the publication standard for Micromachines. 

Author Response

Thank you very much for your comment.

The required figure is redrawn using Visio

Attached is the response
